# Caption-Aware Multimodal Relation Extraction with Mutual Information Maximization

Zefan Zhang
College of Computer Science and Technology, Jilin
University
Key Laboratory of Symbolic Computation and Knowledge
Engineering, Ministry of Education, Jilin University
Changchun, China
zefan23@mails.jlu.edu.cn

Weiqi Zhang
College of Computer Science and Technology, Jilin
University
Key Laboratory of Symbolic Computation and Knowledge
Engineering, Ministry of Education, Jilin University
Changchun, China
zwq23@mails.jlu.edu.cn

Yanhui Li
College of Computer Science and Technology, Jilin
University
Key Laboratory of Symbolic Computation and Knowledge
Engineering, Ministry of Education, Jilin University
Changchun, China
yanhui23@mails.jlu.edu.cn

Tian Bai*
College of Computer Science and Technology, Jilin
University
Key Laboratory of Symbolic Computation and Knowledge
Engineering, Ministry of Education, Jilin University
Changchun, China
baitian@jlu.edu.cn

## Abstract

Multimodal Relation Extraction (MRE) has achieved great improvements. However, modern MRE models are easily affected by irrelevant objects during multimodal alignment which are called error sensitivity issues. The main reason is that visual features are not fully aligned with textual features and the reasoning process may suppress redundant and noisy information at the risk of losing critical information. In light of this, we propose a **C**aption-**A**ware Multimodal Relation Extraction Network with **M**utual **I**nformation **M**aximization (**CAMIM**). Specifically, we first generate detailed image captions through the Large Language Model (LLM). Then, the Caption-Aware Module (CAM) hierarchically aligns the fine-grained visual entities and textual entities for reasoning. In addition, for preserving crucial information within different modalities, we leverage a Mutual Information Maximization method to regulate the multimodal reasoning module. Experiments show that our model outperforms the state-of-the-art MRE models on the benchmark dataset MNRE. Further ablation studies prove the pluggable and effective performance of our Caption-Aware Module and Mutual Information Maximization method. Our code is available at https://github.com/zefanZhang-cn/CAMIM.

## CCS Concepts

• **Computing methodologies** → **Computer vision**.

*The corresponding author.

## Keywords

Multimodal Learning, Multimodal Relation Extraction, Mutual Information Maximization

**ACM Reference Format:**
Zefan Zhang, Weiqi Zhang, Yanhui Li, and Tian Bai. 2024. Caption-Aware Multimodal Relation Extraction with Mutual Information Maximization. In *Proceedings of the 32nd ACM International Conference on Multimedia (MM '24), October 28-November 1, 2024, Melbourne, VIC, Australia.* ACM, New York, NY, USA, 10 pages. https://doi.org/10.1145/3664647.3681219

## 1 Introduction

Relation Extraction (RE) as a significant study in information extraction, aims to detect potential relations among entities in the unstructured text and plays an important role in various applications [12, 32, 39, 43, 50, 59, 60, 62]. Previous studies mostly focus on extracting information from a single textual modality [19, 24–27]. With the popularity of multimodal learning and deep learning [34, 46, 52–54, 66], research abilities that solely on text become limited [65, 67]. Multimodal Relation Extraction (MRE) methods are proposed to significantly assist text-based models by using images as additional inputs [6, 20, 48].

Early methods [3, 33, 35, 61] encode the text through RNN and the image through CNN, studying how to incorporate the feature of the whole image into a text representation. Many works [55, 57, 65] further validate that object-level visual fusion is more specific and important for MRE. Recently, cross-modal pretraining seems promising [18, 42]. Despite the remarkable results achieved by these methods, there are still some challenges:

**Error sensitivity issues.** The multimodal reasoning and fusion process will be interfered with by error sensitivity issues, which means that irrelevant objects are incorporated into textual features and directly harm the multimodal reasoning process. This problem arises from inadequately detailed and accurate descriptions of visual entities during multimodal interaction, leading to the model capturing incorrect entity relations. As shown in Figure 1, to infer

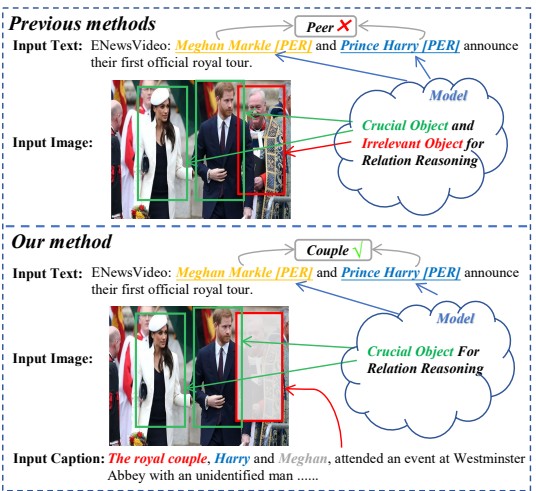

**Figure 1: An example of the Multimodal Relation Extraction.**

the correct relation between "Meghan Markle" and "Harry", previous methods may capture the wrong person due to the lack of detailed descriptions of "Meghan Markle" and "Harry", which may affect the selection of model results.

**Critical information loss issues.** The multimodal reasoning process needs to eliminate the influence of irrelevant image objects and improve the efficiency of relation extraction [6]. However, current methods [3, 6, 33, 35, 61] often overlook the preservation of critical information during multimodal reasoning, potentially leading to the loss of critical information during the suppression of redundant and noisy data.

In our paper, we propose a Caption-Aware Multimodal Relation Extraction Network with Mutual Information Maximization (CAMIM) to achieve the fine-grained multimodal alignment between images and text. Specifically, to tackle error sensitivity issues, the Caption-Aware Module introduces detailed image captions through Multimodal Large Language Model (MLLM) [4], which has the instruction-following ability and the vision-understanding ability in vision and language. Subsequently, we design a cross-attention module to achieve fine-grained multimodal alignment from sentence level and word level. Meanwhile, to tackle the loss of crucial information during multimodal reasoning, we introduce a Mutual Information Maximization method, which is often used as a measure of the correlation between two random variables in information theory. This method preserves task-related crucial information during the multimodal reasoning and supervises different features respectively to ensure the integrity of crucial information. Overall, we summarize the main contributions as follows:

1. We propose a Caption-Aware Multimodal Relation Extraction Network with Mutual Information Maximization (CAMIM). The CAMIM introduces detailed image captions for fine-grained multimodal alignment.

2. We introduce the idea of Mutual Information to preserve crucial information during multimodal reasoning. To the best of our knowledge, this paper is the first work to introduce mutual information into multimodal relation extraction.

3. We evaluate our methods on the MRE dataset and demonstrate their superiority compared to previous state-of-the-art baselines and pluggability.

## 2 Related Work

### 2.1 Multimodal Relation Extraction

Relation Extraction (RE) [7, 31, 58] has gained much attention, recently. Previous studies mainly focus on extracting relations from single text modality [21–23]. Because visual features from images could provide more clues for reasoning, Multimodal Relation Extraction has been proposed and gained more attention. Recently, several studies on multimodal relation extraction aim to utilize relevant images for extracting better relations.

In the early stages, many works [3, 33, 35, 61] propose to encode the text through RNN and encode the image through CNN, then establish the implicit interaction between two modalities. Yu et al. [55], Zhang et al. [57] propose leveraging regional-based image features to represent objects in the image, exploiting fine-grained semantic correspondences based on Transformer. Li et al. [30] propose a fine-grained multimodal alignment approach with Transformer, which aligns visual and textual objects in representation space. Wang et al. [48] propose to retrieve textual evidence from the knowledge base constructed based on Wikipedia. However, most methods ignore the issue of interference from irrelevant objects in the image.

For the multimodal alignment, Sun et al. [42] propose RoBERT to learn a text-image similarity score and filter out the irrelevant visual representations. Chen et al. [6] propose a visual prefix-guided fusion mechanism to remove irrelevant objects. For more fine-grained alignment, Hu et al. [18] propose entity-object and relation-image alignment pretraining tasks to improve MRE performance.

Although these methods make continuous progress, due to the limited description of entities in the text, there is still a problem of being easily affected by irrelevant objects during the process of establishing associations with images, which can affect the selection of model results. Hence, we propose to introduce a detailed image caption to assist the model in completing the alignment operation and facilitate the model's reasoning ability.

### 2.2 Mutual Information

In information theory, Mutual Information (MI) is often used as a measure of the dependency between two random variables. Early methods, Tishby and Zaslavsky et al. [44] first propose the application of information-theoretic objectives in deep neural networks. However, at that time it may not be feasible, and variational inference provides a natural approach to approximate this problem.

To narrow the gap between traditional information theory principles and deep learning, Alemi et al. [1] propose the Variational Information Bottleneck (VIB) framework, which approximates the information bottleneck (IB) constraints and enables the application of information-theoretic objectives to deep neural networks. Since then, Amjad and Geiger et al. [2] and He et al. [15] demonstrate the effectiveness of maximizing mutual information in various contexts. However, it is almost impossible to directly estimate mutual information in high-dimensional space, many studies attempt to approximate the true values using variational bounds [5, 8, 38].

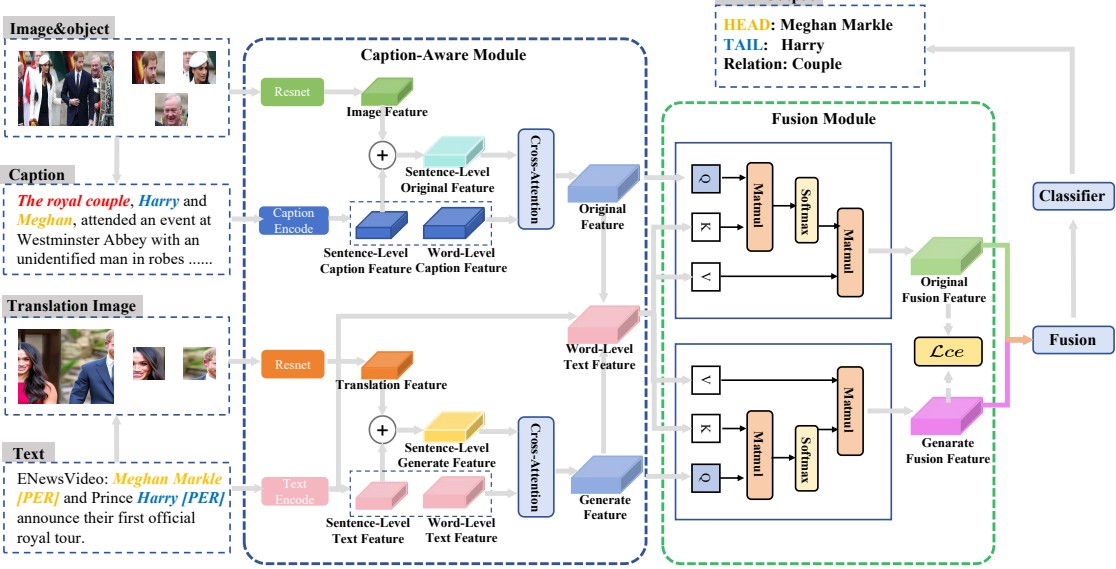

**Figure 2: Caption-Aware Multimodal Relation Extraction Network with Mutual Information Maximization (CAMIM).**

Recently, Han et al. [14] propose a framework that hierarchically maximizes the Mutual Information in unimodal input pairs and between multimodal fusion results and unimodal input in the field of multimodal sentiment analysis. Due to the excellent performance of mutual information in many fields [14, 36, 40, 45], we propose to introduce the idea of mutual information into MRE to maximize the preservation of crucial information during multimodal reasoning.

## 3 Methodology

The Caption-Aware Multimodal Relation Extraction Network with Mutual Information Maximization (CAMIM) is designed to achieve fine-grained multimodal alignment and fusion. The whole framework is shown in Figure 2. Specifically, the Caption-Aware Module incorporates the detailed captions by Multimodal Large Language Model (MLLM) [4] to achieve multimodal alignment from word level and sentence level. In addition, Mutual Information is applied to preserve crucial information. In the following sections, we will provide a detailed description of each module.

### 3.1 Representation of Visual and Textual Features

Firstly, the image contains several visual objects associated with entities in the text, which can provide more semantic knowledge to assist information extraction. Secondly, global image features may express abstract concepts and often serve as weak learning signals. Therefore, we use object-level visual data provided by Chen et al. [6] as a supplement to the global image.

*3.1.1 Visual Features.* We leverage the visual grounding toolkit to extract local visual objects with top m salience [51, 57], and rescale the original image and object images to 224 × 224 followed by Chen et al. [6] as the original images $\mathcal{V}$ and object images $O$.

Meanwhile, we leverage the translation image data generated from textual information provided by Zheng et al. [64], which is also processed through the visual grounding toolkit to obtain the translation original images $\mathcal{V}_T$ and translation object images $O_T$.

As shown in Figure 2, for the multimodal relation extraction task, we input the original images $\mathcal{V}$ and object images $O$ into ResNet50 [16] for encoding and then obtain the original images feature $F_V$ and object images feature $F_O$. We define Image Feature $F_I$ to represent them uniformly, such as Eq.(1):

$$F_I = \{F_V, F_O\}. \tag{1}$$

Similarly, we leverage ResNet50 to encode the translation original images $\mathcal{V}_T$ and translation object images $O_T$, and then obtain the translation original images feature $F_{\mathcal{V}_T}$ and translation object images $F_{O_T}$. We define the Translation Feature $F_T$ to represent them uniformly, such as Eq.(2):

$$F_T = \left\{F_{\mathcal{V}_T}, F_{O_T}\right\}. \tag{2}$$

*3.1.2 Textual Features.* Since the Translation Images are derived from textual data, we directly leverage textual data as a detailed description of the Translation Images. We believe that by concatenating with sentence-level features, the model can learn complete semantic information, thereby achieving coarse-grained alignment. Meanwhile, to learn fine-grained crucial information, we leverage word-level features which can ensure the model selects the correct information between different modalities. Hence, we leverage BERT [11] to encode the text and obtain Sentence-Level Text Feature $T_S$ and Word-Level Text Feature $T_W$.

However, due to the lack of detailed descriptions related to the text in the image, we further leverage a Multimodal Large Language Modal named *QianWen* from Alibaba Cloud and input original

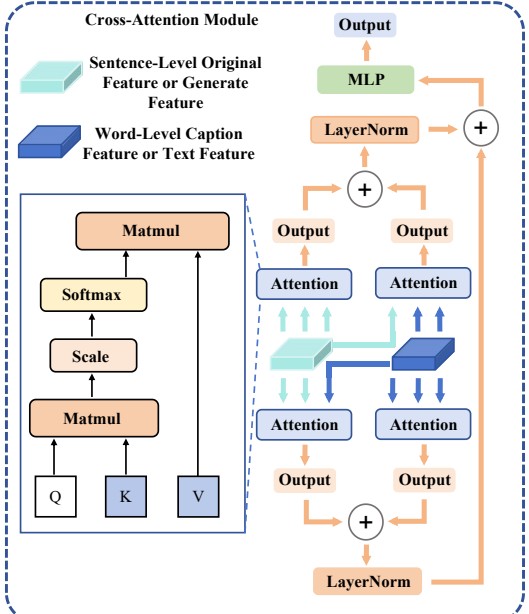

**Figure 3: The illustration of Cross-Attention Module. To achieve better alignment, we first perform self-attention on the different levels of features and then cross attention on them. Then, we concatenate them and process them through normalization and linear layers to obtain the final features.**

image $\mathcal{V}$ to generate a detailed caption $C$ of the whole image to assist the model in completing the alignment operation:

$$prompt = \{\text{Please give me a detailed description of the picture}\},$$
$$\tag{3}$$
$$C = MLLM(prompt, \mathcal{V}), \tag{4}$$

where $MLLM(\cdot)$ denotes the Multimodal Large Language Modal and *prompt* denotes the information that we want to generate. Finally, we also encode caption $C$ using BERT to obtain Sentence-Level Caption Feature $C_S$ and Word-Level Caption Feature $C_W$.

## 3.2 Caption-Aware Module

The main target of the Caption-Aware Module is to achieve better alignment between different modalities. We leverage a two-level alignment method, one is sentence-level feature alignment, and the other is cross-attention-based word-level feature alignment. Sentence-level features are used to align with the overall entity relations in the images, while word-level features are utilized for fine-grained alignment with visual entities.

*3.2.1 **Sentence-Level Features Alignment**.* To achieve alignment between different modalities, we leverage the Multi-Layer Perceptron (MLP) to reduce the dimension of the Image Feature $F_I$, and then concatenate it with the Sentence-Level Caption Feature $C_S$ to obtain the Sentence-Level Original Feature $O_S$:

$$O_S = C_S \oplus (MLP(F_I)), \tag{5}$$

where $MLP$ denotes the MLP layer and $\oplus$ indicates the concatenation operation. Similarly, the Translation Feature $F_T$ is obtained through the MLP layer, which is then concatenated with the $T_S$ to obtain the Sentence-Level Generate Feature $G_S$:

$$G_S = T_S \oplus (MLP(F_T)). \tag{6}$$

By concatenating with sentence-level features, the model learns complete semantic information and achieves coarse-grained alignment. Therefore, we leverage word-level features to learn fine-grained crucial information and achieve better alignment results.

*3.2.2 **Word-Level Features Alignment**.* We design a cross-attention fusion method for the fusion of Sentence-Level Features and Word-Level Features, respectively. As shown in Figure 3, we first perform self-attention on the Sentence-Level Original Feature $O_S$ and the Word-Level Caption Feature $C_W$ separately to obtain $H_S^1$ and $H_W^1$, and then use cross attention to guide the other features to obtain $H_S^2$ and $H_W^2$:

$$SelfAtt(\cdot) = softmax\left(\frac{Q^l \mathcal{K}^l}{\sqrt{d}}\right)\mathcal{V}^l, \tag{7}$$

$$H_S^1 = SelfAtt(O_S), H_W^1 = SelfAtt(C_W), \tag{8}$$

where $Q^l, \mathcal{K}^l, \mathcal{V}^l$ denotes the same level feature as query/key/value.

$$CrossAtt(\cdot) = softmax\left(\frac{Q^m \mathcal{K}^n}{\sqrt{d}}\right)\mathcal{V}^n, \tag{9}$$

$$H_S^2 = CrossAtt(O_S, C_W), H_W^2 = CrossAtt(C_W, O_S), \tag{10}$$

where $m$ and $n$ denote different level features. In $CrossAtt(\cdot)$, the first input is as query, and the second input is as key and value.

Then, we concatenate the different outputs separately and process them through normalization and linear layers to obtain the final Original Feature $O_F$:

$$O_F = MLP_H\left[LN\left(H_S^1 \oplus H_S^2\right) \oplus LN\left(H_W^1 \oplus H_W^2\right)\right], \tag{11}$$

where $LN$ denotes the layer normalization operation and $MLP_H$ denotes the MLP layer.

On the other hand, We leverage the same method to calculate the Sentence-Level Generate Feature $G_S$ and Word-Level Text Feature $T_W$ and obtain the final Generate Feature $G_F$:

$$H_S^3 = SelfAtt(G_S), H_W^3 = SelfAtt(T_W), \tag{12}$$

$$H_S^4 = CrossAtt(G_S, T_W), H_W^4 = CrossAtt(T_W, G_S), \tag{13}$$

$$G_F = MLP_H\left[LN\left(H_S^3 \oplus H_S^4\right) \oplus LN\left(H_W^3 \oplus H_W^4\right)\right]. \tag{14}$$

We believe that fine-grained alignment between different modalities can be achieved through the Caption-Aware Module. Then, we feed the Original Feature $O_F$, Generate Feature $G_F$, and Word-Level Text Feature $T_W$ together into the Fusion Module, ensuring that the model learns crucial task-related information from the text.

## 3.3 Fusion Module

We first perform cross-attention calculations on the Original Feature $O_F$ and Word-Level Text Feature $T_W$ to obtain the Original Fusion Feature $O_{final}$. Then, we use the same method to calculate the Generate Feature $G_F$ and Word-Level Text Feature $T_W$ to obtain the Generate Fusion Feature $G_{final}$:

$$O_{final} = CrossAtt\left(O_S, T_W\right), \tag{15}$$

$$G_{final} = CrossAtt\left(G_S, T_W\right). \tag{16}$$

Finally, we concatenate $O_{final}$ and $G_{final}$ through the MLP layers $MLP_{final}$ and obtain the final fusion feature is $F_{final}$:

$$F_{final} = MLP_{final}\left(O_{final} \oplus G_{final}\right). \tag{17}$$

## 3.4 Classifier

Based on the above description, we obtain the final representation $F_{final}$ and conduct a classifier layer for the relation extraction task. Additionally, we leverage Mutual Information (MI) to capture crucial clues during reasoning.

*3.4.1* **Relation Extraction**. The goal of the relation extraction task is to predict the relation $r$ from the labels $L$ between the subject entity and object entity. Specifically, we leverage a [CLS] head to aggregate the probability distribution over the set of relation labels $L$ with the softmax function. Finally, we calculate the RE loss with the cross-entropy loss function:

$$p(r|X) = softmax(X), \tag{18}$$

$$\mathcal{L}re = -\sum_{i=1}^{n} \log\left(p\left(r|F_{final}\right)\right). \tag{19}$$

*3.4.2* **Mutual Information**. To capture modality-invariant clues between different modalities and ensure the integrity of crucial information related to the task, we leverage MI maximization between $O_{final}$ and $G_{final}$. Inspired by Oord et al. [37] and Han et al. [14], we leverage a score function $s(\cdot)$ that acts on the normalized prediction and truth vectors to gauge their correlation:

$$\overline{O_{final}} = \frac{O_{final}}{\left\|O_{final}\right\|_2}, \overline{G_{final}} = \frac{G_{final}}{\left\|G_{final}\right\|_2}, \tag{20}$$

$$s(O_{final}, G_{final}) = \exp\left(\overline{O_{final}}\left(\overline{G_{final}}\right)^T\right), \tag{21}$$

where $\|\cdot\|_2$ denotes the Euclidean Norm that we obtain unit-length vectors by dividing it because the model intends to stretch both vectors to maximize the score without this normalization. Similarly, we incorporate this score function into the Noise-Contrastive Estimation framework [13] by treating all other representations of that modality in the same batch as negative samples:

$$\mathcal{L}ce = -\mathbb{E}\left[\log \frac{s(O_{final}, G_{final}^i)}{\sum_{G_{final}^j \in G_{final}} s(O_{final}, G_{final}^j)}\right], \tag{22}$$

where $G_{final}^i$ denotes one representation of the Generate Fusion Feature $G_{final}$ in a batch and $G_{final}^j$ denotes other representations of the $G_{final}$ in the same batch.

We ask the Generate Fusion Feature $G_{final}$ to reversely predict representations of the Original Fusion Feature $O_{final}$ to pass more modality-invariant information to $G_{final}$ and improve the performance of the model. Finally, we obtain the loss $\mathcal{L}ce$ between the two features.

Finally, we calculate the weighted sum of all these losses to obtain the main loss for the final:

$$\mathcal{L}r = \mathcal{L}re + \alpha\mathcal{L}ce, \tag{23}$$

where $\alpha$ are hyper-parameter that control the impact of MI maximization, meanwhile, $\mathcal{L}r$ denotes the final loss of the relation extraction task.

## 4 Experiments

In the following section, we conduct experiments to evaluate our method on a multimodal relation extraction task MRE.

## 4.1 Datasets

We evaluate the model on MNRE [65], which contains 12,247 / 1,624 / 1,614 samples in train / dev / test sets, 9,201 images, and 23 relation types. In the case of MRE, a correct extraction of the relation between two entities occurs when the predicted relation type aligns with the gold standard. We adopt Accuracy, Precision, Recall, and F1 as the evaluation metrics. For fair comparisons, our method leverages ResNet50 [16] as the visual backbone and BERT-base [11] as the textual encoder.

## 4.2 Implementation Details

For the textual encoder of our CAMIM model, we leverage the BERT-Base default tokenizer with a max length of 128 to preprocess data. For the visual of our CAMIM model, we leverage ResNet50 to encode the original images and object images from Chen et al. [6] and translation images from Zheng et al. [64].

All optimizations are performed with the AdamW optimizer with a linear warmup of learning rate over the first 10% of gradient updates to a maximum value, then linear decay over the remainder of the training. And weight decay on all non-biased parameters is set to 0.01. We set the number of image objects m to 3.

Remarkably, our model performs based on both HVPNeT [6] and TMR [64], which are previous state-of-the-art methods. On the one hand, we leverage the same settings as HVPNeT, in which we fix the batch size as 32 and learning rates as 3e-5. We train the model for 30 epochs and do an evaluation after the 8th epoch. Moreover, the dimension of the hidden states d is set to 768. The prompt length and prompt dimension remain consistent with the HVPNeT settings. On the other hand, we fix the batch size as 16 and learning rates as 2e-5 which are consistent with TMR. In particular, we train the model for 8 epochs. The dimension of the hidden states d is set to 768. For both HVPNeT and TMR, we leverage the same captions and mutual information methods. Specifically, we set the caption length to 40 and uniformly use a bert-base-uncased encoding. For

**Table 1: Accuracy (%) comparison on MNRE testing set, CAM means the Caption-Aware Module, MI denotes the Mutual Information. TMR and HVPNeT get the best result when using the caption generated by MiniCPM V2.5 and QWenVL-Plus.**

| Methods | MNRE | | | |
|---|---|---|---|---|
| | Accuracy | Precision | Recall | F1 |
| Text Models | | | | |
| BERT (2019) | 74.42 | 58.58 | 60.25 | 59.40 |
| PCNN (2015) | 73.15 | 62.85 | 49.69 | 55.49 |
| MTB (2019) | 75.69 | 64.46 | 57.81 | 60.86 |
| Text+ Image Models | | | | |
| MoRe (2022) | 79.87 | 65.25 | 67.32 | 66.27 |
| MEGA (2021) | 80.05 | 64.51 | 68.44 | 66.41 |
| IFAformer (2023) | 92.38 | 82.59 | 80.78 | 81.67 |
| HVPNeT (2022) | 92.52 | 83.64 | 80.78 | 81.85 |
| TSVFN (2023) | 92.67 | 85.16 | 82.07 | 83.02 |
| MMIB (2024) | - | 83.49 | 82.97 | 83.23 |
| MRE-ISE (2023) | 94.06 | 84.69 | 83.38 | 84.03 |
| MRE (2023) | 93.54 | 85.03 | 84.25 | 84.64 |
| PROMU (2023) | - | 84.95 | 85.76 | 84.86 |
| TMR (2023) | - | 90.48 | 87.66 | 89.05 |
| HVPNeT + MI | 92.13 | 84.96 | 80.31 | 82.57 |
| HVPNeT + CAM | 92.87 | 84.03 | 82.19 | 83.10 |
| HVPNeT + CAM + MI | 93.56 | 85.26 | 84.06 | 84.66 |
| TMR + MI | 94.92 | 91.33 | 87.19 | 89.21 |
| TMR + CAM | 95.35 | 91.04 | 88.91 | 89.96 |
| **TMR + CAM + MI** | **95.79 (+1.73)** | **91.73 (+1.25)** | **90.16 (+2.50)** | **90.94 (+1.89)** |

the calculation of mutual information, we set the hyper-parameter $\alpha$ to 0.1 which is analyzed in 4.5.

## 4.3 Baselines

We compare our method with the following baselines for a comprehensive comparison. The baselines consist of two categories:

**Text-based RE methods** that traditionally leverage merely the texts of MRE data.

• **BERT** [11]: It is the first fine-tuning-based representation model that reduces the need for many heavily-engineered task-specific architectures.

• **PCNN** [56]: It devises a piecewise max pooling layer to capture structural information between different entities.

• **MTB** [41]: It builds task-agnostic relation representations solely from the entity-linked text.

**Multimodal RE methods** that leverage both text and image contents of MRE data.

• **MoRe** [48]: It injects knowledge-aware information into multimodal studies using multimodal retrieval.

• **MEGA** [65]: It employs an efficient alignment strategy for textual and visual graphs to classify textual relations more precisely.

• **IFAformer** [30]: It proposes a method with an implicit fine-grained multimodal alignment based on Transformer.

• **HVPNeT** [6]: It treats visual representations as visual prefixes that can be inserted to guide textual representations of error-insensitive prediction decisions.

• **TSVFN** [63]: It combines the powerful modeling capabilities of graph neural networks and transformers networks to fully fuse critical information between visual and textual modalities.

• **MMIB** [9]: It introduces Information Bottleneck to remove noise in different modalities and aligns multimodal data.

• **MRE-ISE** [49]: It introduces a novel idea of simultaneous information subtraction and addition for multimodal relation extraction.

• **MRE** [20]: It uses cross-modal retrieval for obtaining multimodal evidence to improve prediction accuracy and synthesize visual and textual information for relational reasoning.

• **PROMU** [18]: It enables the extraction of self-supervised signals from massive unlabeled image-caption pairs to pretrain multimodal fusion modules.

• **TMR** [64]: It implements multimodal versions of back-translation and high-resource bridging, which provide a multi-view to the misalignment between modalities.

## 4.4 Main Results

The experimental results of our CAMIM model and all baselines on the MNRE testing set are presented in Table 1. It is easy to see our method outperforms other SOTA methods.

Firstly, we can find that incorporating the visual features is generally helpful for the relation extraction task by comparing the SOTA multimodal approaches with their reliance on pure text-based baselines. Due to the short and ambiguous characteristics of texts in social media, it is difficult to identify entities and their relations in a limited context.

Secondly, our model outperforms HVPNeT and TMR, which leverage hierarchical visual representations or multimodal versions of back-translation and high-resource bridging on the MRE task. The former is a typical baseline in the field of multimodal relation extraction. We add Caption-Aware Module and Mutual Information

**Table 2: Ablation study of different MLLMs and VLMs. CapAcc means the caption accuracy (%), and CapLen means the mean caption length.**

| MLLM | LLM | CapAcc | CapLen | HVPNeT+CAMMI | | | | TMR+CAMMI | | | |
|---|---|---|---|---|---|---|---|---|---|---|---|
| | | | | Accuracy | Precision | Recall | F1 | Accuracy | Precision | Recall | F1 |
| BLIP2 [29] | Flan-T5-XXL | 83.28 | 9 | 92.50 | 83.23 | 82.19 | 82.70 | 94.86 | 89.42 | 89.84 | 89.63 |
| InstructBLIP [10] | Vicuna-7B | 94.10 | 11 | 92.25 | 83.92 | 81.56 | 82.73 | 94.98 | 90.11 | 89.69 | 89.90 |
| LLaVA-NeXT [28] | LLaMA3-8B | 93.11 | 19 | 92.69 | 84.46 | 82.34 | 83.39 | 95.16 | 90.71 | 88.44 | 89.56 |
| QWenVL-Plus [4] | Qwen-LM | 88.63 | 28 | 93.56 | 85.26 | 84.06 | 84.66 | 95.57 | 91.41 | 89.84 | 90.62 |
| CogVLM2 [47] | LLaMA3-8B | 92.90 | 49 | 92.93 | 83.54 | 83.28 | 83.41 | 95.53 | 91.11 | 89.69 | 90.39 |
| MiniCPM V2.5 [17] | LLaMA3-8B | 93.98 | 67 | 92.94 | 84.06 | 84.06 | 84.06 | 95.79 | 91.73 | 90.16 | 90.94 |

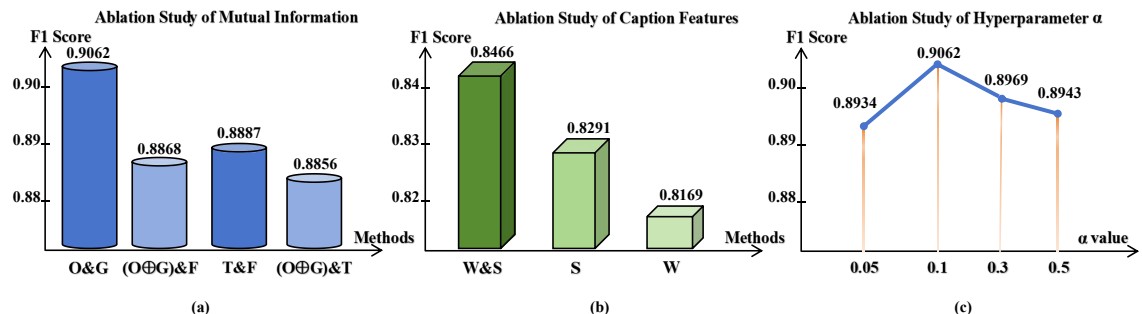

**Figure 4: (a) Ablation study of Mutual Information based on TMR. O: original features, G: generate features, ⊕: concatenation, T: text features, F: final fusion features, *a&b*: MI between a and b. (b) Ablation study of caption alignment method on HVPNeT. Specifically, W&S denote caption alignment methods that leverage both sentence-level features and word-level features, S denotes only leverages sentence-level features, and W denotes only leverages word-level features. (c) Ablation study of Hyperparameter $\alpha$ based on TMR.**

without changing any parameter settings. Specifically, we retain the method of visual prefixes and only add captions to interact with visual features during the prefix processing to achieve better alignment. At the same time, we calculate the mutual information between the final features and visual information to ensure the model preserves crucial information related to the task. As shown in Table 1, compared with the results of HVPNeT, the F1 score increases from 81.85% to 84.66%.

Similarly, we further add the Caption-Aware Module and Mutual Information to the newest SOTA method TMR. We retain the back-translation data processing and align the caption with the original images. At the same time, since the translation images are derived from textual data, we leverage text as the caption of the translation images. Additionally, to ensure the crucial information between the original images and the translation images remains unchanged, we leverage mutual information calculation for them. As shown in Table 1, compared with the results of TMR, the F1 score still increases from 89.05% to 90.94%.

Therefore, the results indicate that our method, which introduces detailed captions of the whole image, and mutual information that can capture modality-invariant clues among modalities, is helpful for relation extraction tasks and can achieve stable and excellent results. Note that our method is pluggable and can be applied to other MRE models.

## 4.5 Ablation Study

In this section, we conduct extensive experiments with the variants of our model to analyze the effectiveness of each component. As shown in Table 1, our baseline methods contain HVPNeT and TMR. Hence, we ablate our Caption-Aware Module and Mutual Information on both of them.

*4.5.1 **Captions generated by different MLLMs**.* We test different captions generated by the six latest MLLMs (LLaVA-NeXT [28], QWenVL-Plus [4], CogVLM2 [47], MiniCPM V2.5 [17], among them, CogVLM2 and MiniCPM V2.5 are open-source multimodal models at the GPT-4V level.) and VLMs (BLIP-2 [29], Instruct-BLIP[10]). We invite graduate students with strong English proficiency to check the accuracy of the generated captions in the test set.

1)The accuracy of the generated captions is shown in Table.2. Due to the poorer language ability of BLIP-2, the accuracy of the generated caption is lower, and there are often some grammar errors. In MLLMs, the most common mistakes are hallucinatory problems. Additionally, the model with the captions generated by MLLMs performs better than those generated by VLMs because of the more detailed captions. Even with a few hallucination mistakes, our model could select the most relevant information related to the head and tail entities.

2)The performance of MLLMs in our models. The captions generated by QwenVL-Plus are of appropriate length and contain substantial content, which has already achieved a notably good effect.

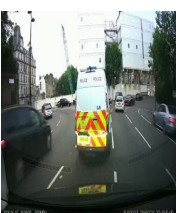

**Text**: @steve_garelick: Why it's best not to cut off the Police in Central London.
Head: Police
Tail: Central London
True: /org/loc/locate_at

w/o MI: None
w/o Caption: None
**CAMIM**: /org/loc/locate_at

**Caption**: A nighttime dashcam view of a busy city street with police van leading traffic, surrounded by various vehicles, buildings, and a crane.

**Case 1**

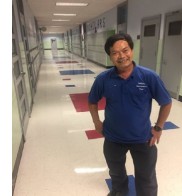

**Text**: @DeMathaCatholic: Among the unsung heroed of DeMatha: Mr.Pham.
Head: Pham
Tail: DeMatha
True: /per/org/member_of

w/o MI: /per/loc/place_of_residence
w/o Caption: /per/per/alternate_names
**CAMIM**: /per/org/member_of

**Caption**: Mr. Pham, a maintenance worker at St. Matha's School, stands confidently in an indoor setting with white walls and blue accents.

**Case 2**

**Figure 5: Several cases predicted by HVPNeT.**

However, MiniCPM V2.5 and CogVLM2 further enrich the captions by incorporating more detailed entities, such as clothing, actions, and behaviors of individuals. The results showed that the impact of the captions generated by MLLMs fluctuated within the normal range. In addition, the TMR+CAMIM with more detailed captions generated by MiniCPM V2.5 achieved the current best performance.

*4.5.2 Caption-Aware Module.* The ablation of Caption-Aware Module. The results in Table 1 show the CAM can boost model performance both in HVPNeT and TMR. It also demonstrates detailed captions of the whole image can assist the model in completing the alignment operation and facilitate inference of correct relation.

Furthermore, as shown in Figure 4 (b), we conduct ablation studies of caption alignment methods on HVPNeT. We can intuitively observe that the method of aligning captions using both word-level features and sentence-level features simultaneously gains better results. However, using any level of feature alone does not gain a satisfactory effect. Therefore, it can be illustrated that both word-level features and sentence-level features contain crucial semantic information and can assist the model in reasoning.

*4.5.3 Mutual Information.* In Table 1, the results indicate that the introduction of mutual information can stably improve the performance on two baselines.

Furthermore, as shown in Figure 4 (a), we conduct ablation studies of mutual information based on TMR (O&G denotes mutual information calculation between the Original features and Generate features, (O⊕G)&F: Visual concatenation result and Final fusion feature, T&F: Text feature and Final fusion feature, (O⊕G)&T: Visual concatenation result and Text feature.). We can observe significant differences in the calculation results of mutual information between different features. The MI result between original features and generated features is the most obvious which means that there may lose some critical information during reasoning. The poor performance of mutual information between different modalities suggests that visual and textual modalities contain different-level semantic information, highlighting the need for joint reasoning after alignment.

Summarily, mutual information can assist the model in capturing modality-invariant clues between different modalities, and the more significant the semantic information differences contained in modalities, the better the ability of the model to capture them. Finally, as shown in Figure 4 (c), we conduct ablation studies of

hyperparameter $\alpha$ in {0.05, 0.1, 0.3, 0.5}, which is used in mutual information calculation. We can easily observe that the best effect is achieved when the value is 0.1.

### 4.6 Case Analysis

We conduct a case study of the Caption-Aware Module (CAM) and Mutual Information Maximization (MI) with HVPNeT, as shown in Figure.5.

In case 1, it is difficult for the model to understand the exacted relation between the head and tail entities, and it is easy to find the correct relation under the guidance of the given caption.

In case 2, the model struggles to identify Dematha's attributes and picks the wrong relation without CAM. The detailed caption, specifically "Mr. Pham, a school maintenance worker at St. Matha's school" guides the model to the right relation. Yet, irrelevant details abound. Lacking MI, the model may get distracted by these irrelevant features. Here, the excessive indoor setting in the caption could mislead the model to choose "place-of-residence" without MI to preserve critical information within different modalities. Concisely, **CAM ensures multi-grained alignment as the model might not comprehend all entities**. Meanwhile, **MI is necessary to retain critical information across modalities since not every caption sentence is relevant**.

## 5 Conclusions

In this paper, to solve the error sensitivity issues and the critical information loss issues, we propose a Caption-Aware Multimodal Relation Extraction Network with Mutual Information Maximization (CAMIM). The Caption-Aware Module introduces detailed image captions generated from the Multimodal Large Language Model (MLLM) to achieve fine-grained alignment between visual and textual information. The Mutual Information Maximization method is designed to preserve crucial information between different modalities. Experiments show that our CAMIM model outperforms the state-of-the-art multimodal relation extraction models.

## Acknowledgments

This work is supported by the National Natural Science Foundation of China [U21A20390], the Development Project of Jilin Province of China [20240601039RC] and the Fundamental Research Funds for the Central University, JLU.

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
