# OpenReview forum: "Caption-Aware Multimodal Relation Extraction with Mutual Information Maximization"
_acmmm.org/ACMMM/2024/Conference — MM2024 Poster_

### Official Review · Reviewer_YjJF · 2024-05-09

**Rating:** 4
**Confidence:** 3

**Summary:**

This paper presents a multimodal relation extraction network that addresses error sensitivity and critical information loss issues. The proposed approach combines a caption-aware module with a mutual information maximization method. Experimental evaluation shows the superiority of the proposed approach compared to previous state-of-the-art methods.

**Strengths:**

1. The primary strength of the method lies in its performance. Experimental evaluation demonstrates its superiority compared to other state-of-the-art methods.

2. The proposed method improves the error sensitivity issues and the critical information loss issues existing in the Multimodal Relation Extraction task.

3. The provided ablation results convincingly showcase the effectiveness of the proposed approach, specifically the caption-aware module and mutual information.

**Limitations:**

1. The case analysis lacks the individual application of the Caption-Aware Module and Mutual Information, resulting in changes.

2. The impact of inappropriate translation image data generated from textual information on the model's performance.

3. In the section 4.2, does "do an evaluation after the 8th epoch" imply selecting the final model based on its performance on the test set?

**Suitability:**

3

---

### Official Review · Reviewer_JkSP · 2024-05-25

**Rating:** 3
**Confidence:** 3

**Summary:**

This paper introduces a method named "Caption-Aware Multimodal Relation Extraction Network with Mutual Information Maximization" (CAMIM), which is tailored to mitigate the challenges of error sensitivity and loss of key information in multimodal relation extraction (MRE). CAMIM harnesses the sophisticated image captions generated by Multimodal Large Language Models (MLLMs) **and incorporates a mutual information maximization approach to facilitate granular alignment and inferencing between visual and textual modalities**. Experimental evidence showcases CAMIM's supremacy over the extant avant-garde MRE models on the MNRE benchmark dataset, with subsequent ablation studies validating the efficacy of its caption-aware component and mutual information maximization strategy.

**Strengths:**

*   This paper conducts valuable research on the impact of irrelevant targets and redundant and noisy information on RE tasks in the context of MRE, making it a valuable research point.
*   The paper demonstrates clear logic and smooth writing.
*   The experimental section of the paper provides comprehensive comparisons with baselines, yielding results that are competitively strong.

**Limitations:**

- The novelty introduced in the paper seems an accumulation of efforts, with enhancing modality alignment through multi-granularity image-text alignment for better downstream task representation always being a focal point in works such as VLMs.
- Where does the advantage of captions generated by LLMs lie over those produced by traditional VLMs? The paper's reference to what it calls MLLM, possibly Qwen-VL, seems to only be described in section 3.1.2, which is rather confusing. Has the author compared the captioning capabilities against currently popular open-source MLLMs? The absence of solid experiments and analysis in this area leaves an impression of trend-chasing.
- In section 3.4, the author employs the concept of Mutual Information Maximization for loss constraint during the classification stage, yet its contribution appears unclear in the ablation study. Claiming this to be the first time mutual information has been introduced to multimodal relation extraction is hard to swallow, as contrastive learning has been extensively researched in MRE and mNER, essentially embodying the core idea of MIM.

**Suitability:**

3

---

### Official Review · Reviewer_Hynp · 2024-05-25

**Rating:** 5
**Confidence:** 3

**Summary:**

This paper introduces a novel approach to Multimodal Relation Extraction (MRE). The proposed CAMIM model addresses critical issues in MRE, such as error sensitivity and critical information loss, by incorporating detailed image captions and employing Mutual Information Maximization. The model utilizes a Caption-Aware Module (CAM) to align fine-grained visual and textual entities and ensures the preservation of crucial information through mutual information techniques. Experiments on the MNRE benchmark dataset show that CAMIM outperforms state-of-the-art models, and ablation studies confirm the effectiveness and pluggability of its components.

**Strengths:**

(1) Introducing detailed image captions generated by a Multimodal Large Language Model (MLLM) for fine-grained multimodal alignment is a novel approach. This method effectively reduces error sensitivity by minimizing the influence of irrelevant objects.

(2) The use of Mutual Information Maximization to preserve crucial information during multimodal reasoning is well-motivated and theoretically sound.

(3) The paper provides extensive experiments demonstrating the superior performance of CAMIM over existing state-of-the-art models on the MNRE dataset. The results are backed by quantitative metrics.

(4) Ablation studies are conducted to validate the individual contributions of the Caption-Aware Module and mutual information techniques, proving their effectiveness and pluggability

**Limitations:**

(1) The effectiveness of the CAMIM model heavily relies on the quality of the generated captions. The paper assumes that the MLLM can consistently produce accurate and detailed captions, but it does not address potential issues arising from incorrect or ambiguous captions. An analysis of the quality of generated captions is urgently required.

(2) Lacks details on the mutual information maximization method and how it benefits the overall performance of the proposed architecture. A case study or visualization may strengthen the contribution of this module.

**Suitability:**

2

---

### Meta-Review · Area_Chair_XNWn · 2024-07-05

**Recommendation:** Accept (Poster)
**Confidence:** 4

**Metareview:**

This paper proposes a Caption-Aware Multimodal Relation Extraction Network with the mutual information maximization technique. They first generate detailed image captions using LLM for fine-grained multimodal alignment between visual entities and textual entities. The mutual information maximization technique is designed to preserve crucial information between different modalities. Experiments on the MNRE dataset validate the effectivess of the method.

(+) On the positive side, the reviewers found the method to be novel and effective and appreciate the clear motivation and good results.

(-) On the negative side, there are still some concerns on the quality of generated captions, insufficient discussion of mutual information maximization, insufficient evaluation on just one datasets and weak presentation.

Several technical questions and suggestions were raised by the reviewers. The authors have taken these into consideration. Overall, all reviewers finally agreed to accept the paper after the rebuttal. Therefore, the AC recommends accepting the paper.